# D$^2$NeRF: Self-Supervised Decoupling of Dynamic and Static Objects from a Monocular Video

**Tianhao Wu**
University of Cambridge

**Fangcheng Zhong**
University of Cambridge

**Andrea Tagliasacchi**
Google Research
Simon Fraser University

**Forrester Cole**
Google Research

**Cengiz Oztireli**
Google Research
University of Cambridge

## Abstract

Given a monocular video, segmenting and decoupling dynamic objects while recovering the static environment is a widely studied problem in machine intelligence. Existing solutions usually approach this problem in the image domain, limiting their performance and understanding of the environment. We introduce Decoupled Dynamic Neural Radiance Field (D$^2$NeRF), a self-supervised approach that takes a monocular video and learns a 3D scene representation which decouples moving objects, including their shadows, from the static background. Our method represents the moving objects and the static background by two separate neural radiance fields with only one allowing for temporal changes. A naive implementation of this approach leads to the dynamic component taking over the static one as the representation of the former is inherently more general and prone to overfitting. To this end, we propose a novel loss to promote correct separation of phenomena. We further propose a shadow field network to detect and decouple dynamically moving shadows. We introduce a new dataset containing various dynamic objects and shadows and demonstrate that our method can achieve better performance than state-of-the-art approaches in decoupling dynamic and static 3D objects, occlusion and shadow removal, and image segmentation for moving objects. Project page: d2nerf.github.io

## 1 Introduction

Reasoning about motion is a fundamental task in machine vision which facilitates intelligent interactions with the 3D environment for applications such as robotics and autonomous driving. Given a monocular RGB video captured from a moving casual camera, we consider the problem of disentangling the camera from object motion, and simultaneously recovering a 3D model of the static environment.

While decomposition of scenes in the image domain has been addressed in the literature, the use of 2D priors and inpainting technique lacks 3D understanding, leading to sub-optimal results. We approach this problem in 3D, aiming to reconstruct a decoupled 3D scene representation that allows for synthesizing the dynamic and static objects *separately* in a free-view and time-varying fashion.

Compared to the task of *static* scene reconstruction [32], modeling a scene with time-dependent effects is a severely ill-posed problem. Existing works seek for robust solutions by incorporating additional supervision such as multi-view capture [25], optical flow [10], or depth [60], but they treat

36th Conference on Neural Information Processing Systems (NeurIPS 2022).

*every* part of the scene as time-dependent, leading to a poor reconstruction of background details due to limited network capacity.

In this paper, we adapt Neural Radiance Fields [32] (NeRF) and its extension HyperNeRF [40] to time-varying scenes by decoupling the dynamic and static components of the scene into separate radiance fields. Previous techniques that decouple dynamic and static scenes either rely on pre-trained object detection/segmentation modules [19, 26, 12, 23], or are limited to a single rigid object [68] or semi-static objects [54]. Our method learns dynamic and static components separately in a self-supervised fashion, using a novel skewed-entropy loss to encourage a clean separation of static and dynamic objects.

A crucial issue in creating a clean separation is *properly handling shadows*, as dynamic objects cast shadows that cause the radiance of the shadow receiver to vary with time. When the shadow receiver is part of the static component, this time-varying change in radiance cannot be directly modeled. Our solution is to relax the static component with a time-varying *shadow field* that modulates the radiance, allowing the shadows cast by moving objects to be captured while constraining density and color to be static.

Our method enables 3D scene decoupling and reconstruction from a monocular video captured from casual equipment such as a mobile phone, and can be readily extended to multi-view videos. By separately modeling the time-varying and time-independent targets in the video, our method can remove the dynamic occluders and their shadows, and synthesize a clean background from novel views.

We demonstrate the effectiveness of our method in *two aspects*: (i) the quality of novel view synthesis of the decoupled static background for monocular videos where the dynamic objects and shadows heavily occlude the scene, and (ii) the correctness of segmentation of dynamic objects and shadows on 2D images.

We introduce a *new dataset* with rigid and non-rigid dynamic objects, rapid camera motion and various moving shadows in both the synthetic and real-world settings to evaluate these two aspects, and show that our method achieves better performance than state-of-the-art approaches.

## 2   Related Work

As our method learns a decoupled neural 3D representation of the dynamic and static scenes, we start this section with a review of scene representations, and then focus on methods for object motion decoupling. We also review prior works for 2D segmentation of moving objects.

**Scene Representations**   A 3D scene representation is a data structure that encodes the geometry and appearance of a 3D scene, upon which many algorithms and applications are developed. Recently, there has been a surge of methods that combine deep learning methods with traditional 3D representations: point clouds [17, 41, 11, 1], meshes [7, 33, 3, 55], voxels [6, 59], implicit surfaces [38, 63, 15, 14, 51], light fields [50, 52, 2, 48] and volumetric fields [28, 32]. Among neural representations, NeRF [32] has attracted substantial attention due to its photo-realistic performance in novel view synthesis for scenes with complex geometry, lighting, and materials. Via differentiable volume rendering and inputs of multiple views of the scene, NeRF applies an MLP to learn a 5D radiance field of the scene modeling the spatially and view-dependent radiance. Various extensions of NeRF have been developed to improve its performance and generality such as training with only one or few views [67, 18, 22, 35, 45], allowing for input images with inconsistent lighting and object locations [31, 69], learning large-scale scenes with street or satellite views [47, 61], speeding up rendering to reduce training and inference time [29, 8, 44, 46, 66, 13, 34, 27, 53, 58, 65], capture of dynamic effects within the scene [25, 43, 10, 60, 39, 40, 21, 36], and decomposing self-occluded shadows to recover correct albedo [9]. We further extend NeRF to *decouple* dynamic from static effects.

**Motion Decoupling**   Prior works to acquire a decoupled 3D representation of dynamic and static scenes can be divided into either supervised or self-supervised approaches. Among the supervised approaches, STNeRF [19] learns individual NeRFs with deformation fields for each human in a dynamic scene through pre-trained human segmentation networks. Similarly, NSFF [26] and DynNeRF [12] rely on pre-trained semantic and motion segmentation methods to obtain masks for

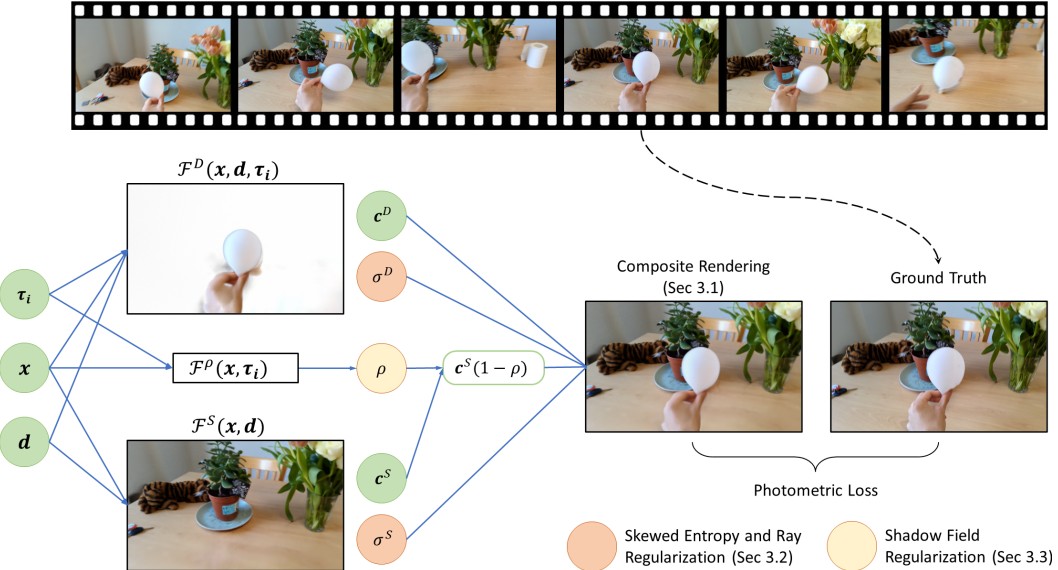

Figure 1: **Overview** – Given the ground truth view, camera pose and the time frame, our method reconstructs the underlying scene as a composite radiance field. Dynamic objects are represented by $\mathcal{F}^D$, while the static scene is represented by $\mathcal{F}^S$. The shadow-field $\mathcal{F}^\rho$ models non-static shadows within the input video.

moving objects in a monocular video, and explicitly guide the training of separate NeRF networks to decouple the scene based on motion. Among the self-supervised approaches, SIMONe [20] incorporates a transformer encoder and variational autodecoder to simultaneously recover novel views, object segmentation masks and dynamic object trajectories, but they do not allow for a synthesis of dynamic or static objects alone. NeRF-W [31] employs per-frame embeddings to model non-photometric consistent effects in unconstrained photo collections, but their design was not intended for a clean separation between moving objects and the static scene. STaR [68] reconstructs and decomposes a rigid dynamic object and the static background simultaneously by optimizing two NeRFs and a set of time-varying object poses in a self-supervised way, but it is only suitable for scenes with a single rigid dynamic object and requires multi-view videos. Conversely, our approach works with more complex scenes involving multiple non-rigid and topologically varying objects, and our method can be directly applied from monocular video. NeuralDiff [54] incorporates three NeRF-based streamlines to decompose background, object and actor from an egocentric video, and it is the most similar to ours within the literature. However, its use of a naive time-varying NeRF architecture leads to blurry results and therefore heavily limits its performance on both scene decomposition and reconstruction.

**Image Segmentation of Moving Objects**    Orthogonal to the reconstruction and disentanglement in 3D, there have also been extensive researches in self-supervised and template-free segmentation at the image level (i.e. 2D). The majority relies on motion-clues to segment objects with different optical flow patterns [4, 64, 37, 62]. Some techniques incorporate a transformer style slot-based attention scheme to learn consistent object segmentation over a sequence of optical flow images [64], while others learn alpha-matting from a single video with smooth camera movement and homographic background and extend the segmentation target to correlated effects such as shadow or reflectance [30]. Those approaches come with obvious shortcomings, as they focus on *image* level segmentation and incorporate no 3D understanding, they cannot handle large scale camera motion, complicated static background and cannot recover 3D geometry, or perform novel view synthesis.

## 3    Method

Given a *monocular* video captured from a freely *moving* hand-held camera, our method reconstructs a neural scene representation that decouples moving objects from the static environment, assuming a

constant illumination and known camera poses (e.g. calibrated with COLMAP [49]). As illustrated in Figure 1, our method achieves this by learning *separate* radiance fields for static and dynamic portions of the scene, and doing so in a fully self-supervised fashion. We describe our architecture (Section 3.1), detail of our self-supervised losses (Section 3.2), and describe how, while shadows are not explicitly modeled by NeRFs, a simple technique for their effective removal is attainable (Section 3.3).

## 3.1 Composite Neural Radiance Field

The static component builds upon NeRF [32], which represents the scene as continuous spatial-dependent density $\sigma$ and spatial-view-dependent radiance $\mathbf{c}$ using an multi-layer perceptron $\mathcal{F}^S$:

$$\left.\begin{array}{l} \sigma^S(\mathbf{x}) \in \mathbb{R} \\ \mathbf{c}^S(\mathbf{x}, \mathbf{d}) \in \mathbb{R}^3 \end{array}\right\} = \mathcal{F}^S(\mathbf{x}, \mathbf{d}) \tag{1}$$

where $\mathbf{x} \in \mathbb{R}^3$ is the spatial coordinate, and $\mathbf{d} \in \mathbb{R}^3, \|\mathbf{d}\| = 1$ is the view direction. To model the dynamic component of a scene, we adapt HyperNeRF [40], which accurately captures scenes with *non-rigid* motion as well as *topological changes* by introducing additional degree of freedom and network capacity. For convenience, we denote it as a neural function $\mathcal{F}^D$:

$$\left.\begin{array}{l} \sigma^D(\mathbf{x}, \boldsymbol{\tau}_i) \in \mathbb{R} \\ \mathbf{c}^D(\mathbf{x}, \mathbf{d}, \boldsymbol{\tau}_i) \in \mathbb{R}^3 \end{array}\right\} = \mathcal{F}^D(\mathbf{x}, \mathbf{d}, \boldsymbol{\tau}_i) \tag{2}$$

where $\boldsymbol{\tau}_i \in \mathbb{R}^m$ is the per-frame time latent code. Given a camera ray $\mathbf{r} = \mathbf{o} + t\mathbf{d}$ originating from $\mathbf{o}$ and with direction $\mathbf{d}$, the two models are then composited to calculate the color $\hat{C}$ of the camera ray by integrating the radiance according to volumetric rendering within a pre-defined depth range $[t_n, t_f]$:

$$\hat{C}(\mathbf{r}, \boldsymbol{\tau}_i) = \int_{t_n}^{t_f} T(t) \left( \sigma^S(t) \cdot \mathbf{c}^S(t) + \sigma^D(t, \boldsymbol{\tau}_i) \cdot \mathbf{c}^D(t, \boldsymbol{\tau}_i) \right) dt \tag{3}$$

$$T(t) = exp \left( - \int_{t_n}^{t} (\sigma^S(s) + \sigma^D(s, \boldsymbol{\tau}_i)) ds \right) \tag{4}$$

where we simplify our notation as $\sigma(t) \equiv \sigma(\mathbf{r}(t))$ and $\mathbf{c}(t) \equiv \mathbf{c}(\mathbf{r}(t), \mathbf{d})$. Note that, with such an additive decomposition, samples from either fields are capable of terminating the camera ray and occluding the other.

## 3.2 Supervision Losses

To find the parameters of the static (Eq. 1) and dynamic (Eq. 2) NeRF networks, a photometric loss is applied to ensure that the output image sequences of the composite NeRF (Eq. 3) align with the input video frames:

$$\mathcal{L}_p(\mathbf{r}, \boldsymbol{\tau}_i) = \|\hat{C}(\mathbf{r}, \boldsymbol{\tau}_i) - C(\mathbf{r}, \boldsymbol{\tau}_i)\|_2^2 \tag{5}$$

where $C(\mathbf{r}, \boldsymbol{\tau}_i)$ indicates the true color of camera ray $\mathbf{r}$ obtained from the $i$-th input video frame. However, note the dynamic component can naturally take over the static counterpart by incorrectly assigning occupancy of static objects to dynamic NeRF, and the photometric loss alone also does not guarantee a correct separation. In what follows, we design a collection of regularizers that promote such decoupling in a self-supervised fashion.

**Dynamic vs. Static Factorization** As physical objects cannot co-exist at the same spatial location, a physically realistic solution should have any position in space *either* occupied by a the static scene or by a dynamic object, but *not both*. To enforce this behavior we denote the spatial ratio of dynamic vs. static density as:

$$w(\mathbf{x}, \boldsymbol{\tau}_i) = \frac{\sigma^D(\mathbf{x}, \boldsymbol{\tau}_i)}{\sigma^D(\mathbf{x}, \boldsymbol{\tau}_i) + \sigma^S(\mathbf{x})} \in [0, 1] \tag{6}$$

and then penalize its deviation from a categorical $\{0, 1\}$ distribution via a binary entropy loss [68]:

$$\mathcal{L}_b(\mathbf{r}, \boldsymbol{\tau}_i) = \int_{t_n}^{t_f} H_b(w(\mathbf{r}(t), \boldsymbol{\tau}_i)) dt \tag{7}$$

$$H_b(x) = -(x \cdot log(x) + (1 - x) \cdot log(1 - x)) \tag{8}$$

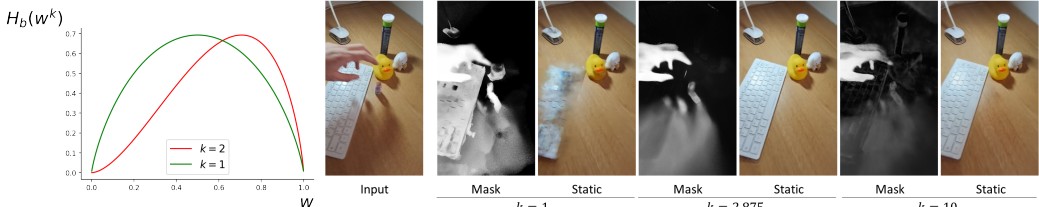

Figure 2: **Skewed entropy** – (left) the skewed ($k > 1$) and classical ($k = 1$) entropy losses. A skewed entropy encourages a wider range of $w$ to decrease and has a larger gradient on values around $0.5$, but its gradient vanishes when $w$ approaches $0$. (right) The decoupled alpha masks and static components when original, properly-skewed and over-skewed binary entropy losses are applied.

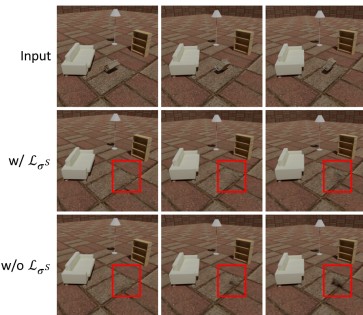

Figure 3: **Static regularization** – By encouraging a more concentrated density distribution along each camera ray in static component, the recovered background contains less view-dependent artifacts.

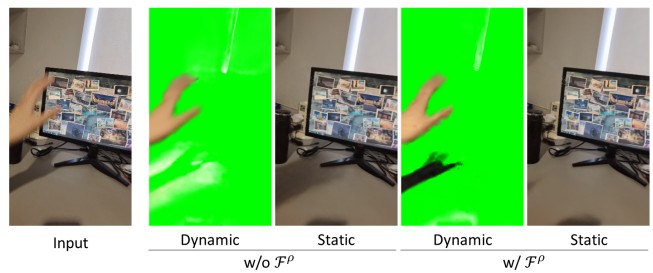

Figure 4: **Shadow ambiguity –** When shadows occur frequently in the input data, the average shadow gets integrated in the static component, and the dynamic component incorrectly learns the differential with respect to this average and appears as a brighter surface. This can be avoided by a more direct modeling of shadow effects as a dynamic darkening of static regions (i.e. the shadow field).

However, due to the strong expressive power of the dynamic networks (Eq. 2), optimizing the loss (Eq. 7) leads to the technique modeling parts of the scene as dynamic, regardless of whether they are dynamic or static; see Figure 2 (right). To overcome this issue, we propose a *skewed* entropy loss to bias our loss to *slightly favor* static explanations of the scene with skewness hyper-parameter $k$, that, as illustrated in Figure 2 (left, $k>1$), attains the desired behavior:

$$\mathcal{L}_s(\mathbf{r}, \boldsymbol{\tau}_i) = \int_{t_n}^{t_f} H_b(w(\mathbf{r}(t), \boldsymbol{\tau}_i)^k) \, dt \tag{9}$$

**Ray Regularization**   Choosing a large value of skewness $k$ causes the appearance of fuzzy floaters (low-density particles) in the static portion of the scene; see Figure 2 (right, $k$=10). As it can be intuitively understood from Figure 2 (left), this is caused by the small gradients of $H_b(x^k)$ as $x$ approaches zero. To mitigate this effect, and reduce fuzziness in the reconstruction, we penalize the maximum of $w$ along each camera ray:

$$\mathcal{L}_r(\mathbf{r}, \boldsymbol{\tau}_i) = \max_{t \in [t_n, t_f]} w(\mathbf{r}(t), \boldsymbol{\tau}_i) \tag{10}$$

Such loss can be intuitively interpreted as constraining the dynamic component to occupy as few pixels as possible while keeping minimal impact on the overall loss for all samples. Note that $\mathcal{L}_r$ only removes density floaters that sit along camera rays that *do not intersect* with any dynamic objects.

**Static Regularization**   We empirically found that static component may abuse the camera pose as the hint for the current time frame and learn dynamic effects as sparse clouds that lead to high-frequency appearance changes; see Figure 3. The ambiguity comes from the fact that we are using monocular casual videos where the camera almost never visits the exact same position twice during the capture. That is, there exists a one-to-one mapping between camera pose and time variable. We

solve this issue by imposing a prior on the distribution of density along a ray, penalizing density distributions that would cause cloud-like artifacts [22, 45]:

$$\mathcal{L}_{\sigma^S}(\mathbf{r}) = -\int_{t_n}^{t_f} p(t) \cdot log \, p(t) \, dt \quad \text{where} \quad p(t) = \frac{\sigma^S(\mathbf{r}(t))}{\int_{t_n}^{t_f} \sigma^S(\mathbf{r}(s)) \, ds} \tag{11}$$

### 3.3 Shadow Fields

Neural radiance fields cannot faithfully model standalone shadows without significant changes to its architecture necessary to modeling materials and illumination; see NeRFactor [71]. In simple cases where shadows of the dynamic objects move rapidly, they could alternatively be learned by the dynamic radiance field as semi-transparent layers on top of the static surface. However, this tends to fail for shadows that do not move much, or that are highly correlated with the camera motion. As shadows are texture-less, understanding their movement is ambiguous, and representing them as a semi-transparent layer causes difficulties in the optimization; see Figure 4. To overcome this issue, and under the assumption of a direct illumination model (i.e. negligible global illumination effects), we make the observation that a cast shadow can be represented as a *pointwise reduction* in the radiance of the the static scene, and incorporate this within Eq. 3 as:

$$\hat{C}(\mathbf{r}, \boldsymbol{\tau}_i) = \int_{t_n}^{t_f} T(t)((1 - \underbrace{\rho(\mathbf{r}(t), \boldsymbol{\tau}_i)}) \cdot \sigma^S(t) \cdot \mathbf{c}^S(t) + \sigma^D(t, \boldsymbol{\tau}_i) \cdot \mathbf{c}^D(t, \boldsymbol{\tau}_i)) \, dt \tag{12}$$

$$\rho(\mathbf{x}, \boldsymbol{\tau}_i) \in [0, 1] = \mathcal{F}^\rho(\mathbf{x}, \boldsymbol{\tau}_i) \tag{13}$$

where $\rho(\mathbf{x}, \boldsymbol{\tau}_i)$ is a *shadow ratio* that scales-down the radiance of the static scene to incorporate the shadow. To avoid the shadow-ratio from over-explaining dark regions of the scene, we penalize its average squared magnitude along a ray:

$$\mathcal{L}_\rho(\mathbf{r}, \boldsymbol{\tau}_i) = \frac{1}{t_f - t_n} \int_{t_n}^{t_f} \rho(\mathbf{r}(t), \boldsymbol{\tau}_i)^2 \, dt \tag{14}$$

Finally, note that shadows cast from dynamic objects onto other dynamic objects are *already* expressed from the radiance term of the dynamic branch, and do not need explicit modeling.

## 4 Experiments

### 4.1 Implementation details

Our method is easily reproducible, as we intend to release code and datasets upon publication to facilitate future research. We adopt the HyperNeRF [40] architecture as the dynamic component, which has a NeRF MLP network of 8 layers, each with 256 channels, and our static NeRF component has the same architecture. Similar to NeRF [32], we apply a hierarchical volume sampling with 64 coarse and 64 fine samples. The optimization takes $100k$ iterations with batch size 1024 and an exponentially decayed learning rate from $10^{-3}$ to $10^{-5}$. This training procedure spans approximately two hours on four NVIDIA A100-SXM-80GB GPUs. The overall loss of our method is:

$$\mathcal{L}(\mathbf{r}, \boldsymbol{\tau}_i) = \mathcal{L}_p(\mathbf{r}, \boldsymbol{\tau}_i) + \lambda_s \mathcal{L}_s(\mathbf{r}, \boldsymbol{\tau}_i) + \lambda_r \mathcal{L}_r(\mathbf{r}, \boldsymbol{\tau}_i) + \lambda_{\sigma^S} \mathcal{L}_{\sigma^S}(\mathbf{r}) + \lambda_\rho \mathcal{L}_\rho(\mathbf{r}, \boldsymbol{\tau}_i) \tag{15}$$

where $\lambda_s$, $\lambda_r$, $\lambda_{\sigma^S}$, $\lambda_\rho$ are the weights of the regularization terms respectively. For scenes with a mixture of dynamic objects and shadows, we apply shadow decay and set $\lambda_\rho$=0.1. We set $\lambda_\rho$=0.001 for scenes featuring view-correlated dynamic shadows only. We experimentally found that the optimal choice of the hyperparameters, especially $\lambda_b$, $\lambda_r$ and the skewness $k$, are strongly influenced by the level of object motion, camera motion, and video length. Therefore, we performed a grid search on our synthetic and held-out real-world scenes, and some scenes from DAVIS [42], to establish a set of hyperparameters applicable to a variety of scenarios; details about hyperparameters can be found in the supplementary. We do not apply shadow field for evaluations on our synthetic scenes, as we empirically found that shadow field is not needed to learn correct shadows. We also disable the view direction input for synthetic scenes as they do not contain strong view-dependent effects.

| | Pick2 | Duck | Balloon | Water | Cookie | Mean |
|---|---|---|---|---|---|---|
| NeuralDiff [54] | **.208** | .222 | .167 | .172 | .159 | **.186** |
| HN [40] | .486 | .253 | .187 | .361 | .162 | .290 |
| Ours | .253 | **.214** | **.153** | **.153** | .156 | **.186** |

| | Car | Cars | Bag | Chairs | Pillow | Mean |
|---|---|---|---|---|---|---|
| MG [64] | .603 | .363 | .629 | .484 | .044 | .424 |
| NeuralDiff [54] | .806 | .508 | .080 | .368 | .097 | .372 |
| Ours | **.848** | **.790** | **.703** | **.551** | **.693** | **.717** |

NeuralDiff    HyperNeRF    Ours

Figure 5: **Novel view synthesis** – LPIPS$\downarrow$

MG    NeuralDiff    Ours

Figure 6: **Video segmentation** – $\mathcal{J}\uparrow$

## 4.2 Evaluation

We demonstrate the performance of our method both quantitatively and qualitatively on three tasks. We focus our attention to our main objective of decoupling and removing dynamic objects, including their shadows, with a 3D reconstruction of the static environment. We only include a summary of our results for novel view synthesis (in Figure 5) and video segmentation (in Figure 6), and refer the reader to the Supplementary E and D for a more in-depth discussion. We strongly encourage the readers to watch videos on the project page (https://d2nerf.github.io/) to better appreciate our results.

## 4.3 Datasets

In addition to the data obtained from HyperNeRF [40] and Nerfies [39], we acquire more complex datasets in the real-world, as well as design a synthetic dataset to enable quantitative comparisons.

**Synthetic dataset** We generate a synthetic dataset with ground-truth masks for moving objects and their shadows with Kubric [16]. This dataset consists of five scenes containing one or multiple dynamic objects from ShapeNet [5] with rigid or non-rigid motion, and the corresponding Kubric worker script is provided in our supplementary material. We move the virtual camera over 10 keyframes randomly sampled from azimuth $[2, 2 + \pi/4]$ and altitude $[1, 1.2]$ to generate a 200-frame video sequence for training. We also rotate the virtual camera around the center of all keyframes to generate 100 validation views with only the static background being visible. We additionally generate masks for both the dynamic objects and their shadows, allowing us to quantitatively study the performance of our algorithm. Note that shadows are usually absent in existing moving-objects segmentation benchmarks.

**Real-world dataset** We also capture ten video sequences of real scenes to showcase our performance. Compared to HyperNeRF's, our dataset contains more challenging scenarios with rapid motion and non-trivial dynamic shadows. Note however that we *cannot* perform quantitative analysis for these datasets due to the absence of ground-truth views of a static scene or ground-truth masks. Five of real scenes are captured with a similar setting as Nerfies, where we use a dual-hold phone rig and synchronize the capture based on audio. We use the images captured by one of the two phones as validation views for novel-view synthesis, which are discussed in the Supplementary E. To demonstrate the ability of fully self-supervised scene decoupling, we *do not* apply any masks when registering real-world images using COLMAP [49].

## 4.4 Scene Decoupling – Table 1, Figure 7

We report the evaluation of our method on its ability to decouple dynamic objects and their shadows, while recovering the static background. We evaluate our performance against NeRF-W [31], NSFF [26] and NeuralDiff [54]. For NeRF-W, we used only transient embedding and disabled the appearance embedding that models variable lighting for evaluation on synthetic scenes, as they have constant illumination. For NSFF, we disabled the hard mining initialization via 2D masks input. For NeuralDiff, we disabled the actor component (as our input videos are not egocentric) and only use the transient component.

| | Car | | | Cars | | | Bag | | | Chairs | | | Pillow | | | Mean | | |
|---|---|---|---|---|---|---|---|---|---|---|---|---|---|---|---|---|---|---|
| | LPIPS↓ | MS-SSIM↑ | PSNR↑ | LPIPS↓ | MS-SSIM↑ | PSNR↑ | LPIPS↓ | MS-SSIM↑ | PSNR↑ | LPIPS↓ | MS-SSIM↑ | PSNR↑ | LPIPS↓ | MS-SSIM↑ | PSNR↑ | LPIPS↓ | MS-SSIM↑ | PSNR↑ |
| NeRF-W [31] | .218 | .814 | 24.23 | .243 | .873 | 24.51 | .139 | .791 | 20.65 | .150 | .681 | 23.77 | .088 | .935 | 28.24 | .167 | .819 | 24.28 |
| NSFF [26] | .200 | .806 | 24.90 | .620 | .376 | 10.29 | .108 | .892 | 25.62 | .682 | .284 | 12.82 | .782 | .343 | 4.55 | .478 | .540 | 15.64 |
| NeuralDiff [54] | .065 | .952 | 31.89 | .098 | .921 | 25.93 | .117 | .910 | 29.02 | .112 | **.722** | 24.42 | .565 | .652 | 20.09 | .191 | .831 | 26.27 |
| Ours | **.062** | **.975** | **34.27** | **.090** | **.953** | **26.27** | **.076** | **.979** | **34.14** | **.095** | .707 | **24.63** | **.076** | **.979** | **36.58** | **.080** | **.919** | **31.18** |

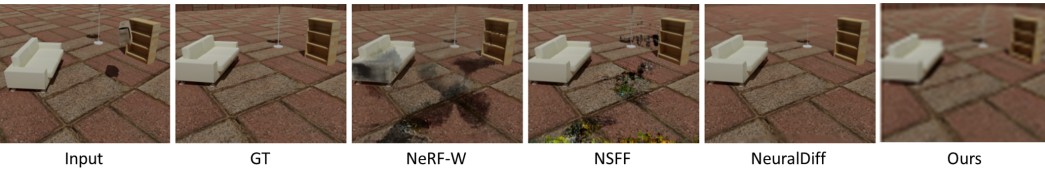

Table 1: **Scene decoupling (quantitative) –** We train on each scene with 200 frames, decouple the dynamic objects and shadows, and render the static component from 100 novel views to compare with ground truth. Note these are computed on the *synthetic* dataset, for which ground truth is available.

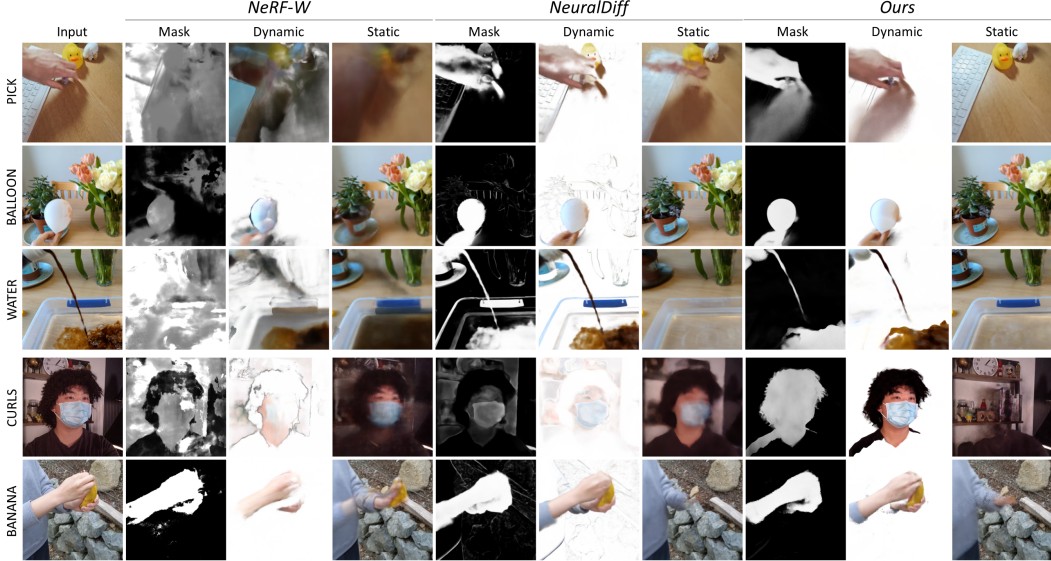

Figure 7: **Scene decoupling (qualitative)** – We visualize results on (top) our new real scenes, and on (bottom) scenes from HyperNeRF [40] and Nerfies [39]. To better illustrate the decoupled object and shadow, we render the dynamic component with a white background. Note that since our method only decouples dynamic targets, it does not include parts of objects that remain still throughout the capture, such as the body in "curls" and "banana" scenes.

In the evaluation, we used each method to synthesize the static background from multiple validation views with the moving objects and their shadows removed; see Figure 7 for qualitative results on real data, as well as Supplementary C for qualitative results on synthetic data. We compare the results with the ground truth on the synthetic data and report LPIPS [70], Multi-Scale SSIM [57], and PSNR as the metrics for novel view synthesis of the decoupled static background; see Table 1. Note that although NSFF similarly learns separate dynamic and static NeRFs, it is not designed to maximize the performance of scene decoupling and does not incorporate regularization on the dynamic NeRF. Therefore, it is extremely unstable and can fail completely by learning everything as dynamic, leaving static component as an empty scene.

## 4.5  Ablations – Table 2, Figure 8, Figure 9

We quantitatively ablate our method on our synthetic dataset; see Table 2: where "skew" means skewness is applied in the binary entropy regularization, and "$\mathcal{L}_r$" stands for the density ratio ray regularization. We also qualitatively ablate it on a real scene; see Figure 8. We also qualitatively illustrate the ablation on shadow field network, which is necessary for decoupling shadows with

| skew | $\mathcal{L}_r$ | Car | | | Cars | | | Bag | | | Chairs | | | Pillow | | | Mean | | |
|---|---|---|---|---|---|---|---|---|---|---|---|---|---|---|---|---|---|---|---|
| | | LPIPS↓ | MS-SSIM↑ | PNSR↑ | LPIPS↓ | MS-SSIM↑ | PNSR↑ | LPIPS↓ | MS-SSIM↑ | PNSR↑ | LPIPS↓ | MS-SSIM↑ | PNSR↑ | LPIPS↓ | MS-SSIM↑ | PNSR↑ | LPIPS↓ | MS-SSIM↑ | PNSR↑ |
| ☐ | ☐ | .214 | .834 | 26.26 | .119 | .943 | 26.10 | .254 | .666 | 19.96 | .104 | .698 | 24.42 | .385 | .671 | 14.24 | .215 | .762 | 22.20 |
| ☑ | ☐ | .182 | .865 | 25.89 | .260 | .803 | 22.47 | .189 | .893 | 28.38 | .107 | .693 | 24.44 | .311 | .770 | 15.27 | .210 | .805 | 23.29 |
| ☐ | ☑ | .067 | .973 | 34.06 | .104 | .948 | 26.19 | .091 | .955 | 31.55 | .151 | .653 | 22.92 | .118 | .940 | 28.17 | .106 | .894 | 28.58 |
| ☑ | ☑ | **.062** | **.975** | **34.27** | **.090** | **.953** | **26.27** | **.076** | **.979** | **34.14** | **.095** | .707 | **24.63** | **.076** | **.979** | **36.58** | **.080** | **.919** | **31.18** |

Table 2: **Ablations (quantitative)** – We train on each scene with 200 frames, decouple the dynamic objects and shadows, and render the static component from 100 novel views for metric evaluations.

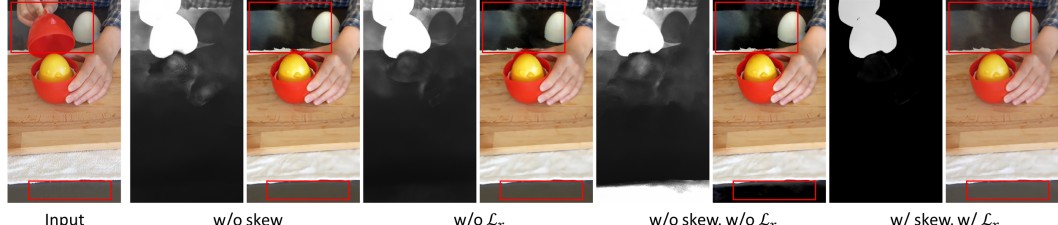

| Input | w/o skew | w/o $\mathcal{L}_r$ | w/o skew, w/o $\mathcal{L}_r$ | w/ skew, w/ $\mathcal{L}_r$ |

Figure 8: **Ablations (qualitative) –** For scenes with slow motion or strong view-dependent reflectance, $\mathcal{L}_r$ is used together with the skewed entropy to prevent the dynamic component from incorrectly decoupling the scene. In the scene above this appears as a slightly darkened color on the table.

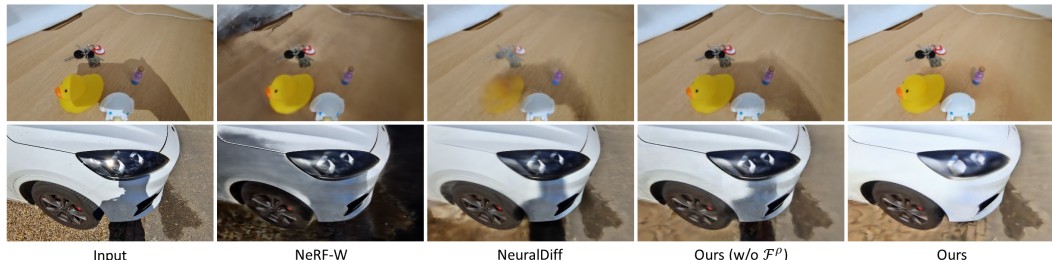

| Input | NeRF-W | NeuralDiff | Ours (w/o $\mathcal{F}^\rho$) | Ours |

Figure 9: **Ablations (shadows) –** Our method is able to remove large area of shadows, even if they are strongly correlated with the view direction (e.g., shadow cast by camera or the photographer). Note that the appearance embedding from NeRF-W [31] is not sufficient to remove shadow that is present throughout the capture; see the our website for additional qualitative results.

large area, slow or repetitive motion, or shadows that are highly correlated with the camera view; see Figure 9.

## 5 Conclusions

We presented D²NeRF, a method for self-supervised 3D scene decoupling and reconstruction from casual monocular videos. Our method decouples occluders and correlated shadows, recovers clean background representations, and enables high quality novel views synthesis. Our novel skewed entropy regularizer is critical to separate dynamic from static components of the scene, while our shadow-field allows for the removal of dynamic shadows without having to explicitly model the interaction between light and geometry. We demonstrate superior results for multiple tasks on existing datasets, as well as on two datasets that we introduce alongside our technique.

**Limitations** Similar to many NeRF-based methods, our approach relies on accurate camera registration to achieve success decoupling and reconstruction. Our approach also suffers from high frequency view-dependent radiance change, such as those caused by the presence of reflective surface within the scene. Due to the monocular moving camera setting, those effects could be misinterpreted as dynamic effects, resulting in incorrect decoupling. Removing texture-less target that repeatedly moves within a very small range is also difficult, as the motion clues are extremely ambiguous in this case; see Figure 10. We also note that the skewed binary entropy is not the only loss that suits the purpose of separating two NeRF components while suppressing one of them. Many losses such as the beta distribution from [28] and Laplacian distribution from [45] can behave similarly by

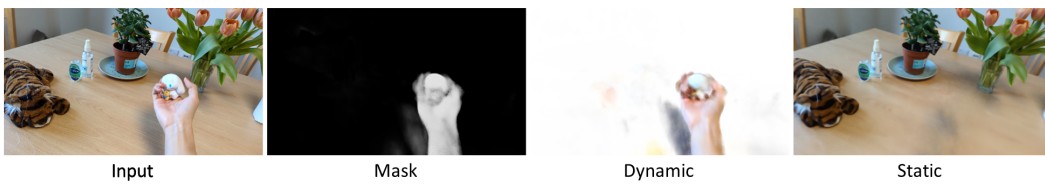

| Input | Mask | Dynamic | Static |

Figure 10: **Limitations –** Because the hand stays around the same position for the majority of the time during the video, our method is unable to fully decouple and remove the texture-less shadow.

incorporating an exponential coefficient. The main difference between our skewed entropy loss and beta or Laplacian distributions is that the former has a vanishing gradient when approaching 0, while the gradients of the latter become infinity. Choosing an appropriate form of loss could reduce the complexity of hyperparameters without affecting the decoupling accuracy.

## Acknowledgements

This work was supported by a UKRI Future Leaders Fellowship [grant number G104084]. We thank Ziwen Cui and Ningding Wang for their help with data collection.

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
