# OpenReview forum: "D^2NeRF: Self-Supervised Decoupling of Dynamic and Static Objects from a Monocular Video"
_NeurIPS.cc/2022/Conference — NeurIPS 2022 Accept_

### Official Review · Reviewer_PH91 · 2022-07-05

**Rating:** 5
**Confidence:** 4
**Soundness:** 3 good
**Presentation:** 3 good
**Contribution:** 2 fair

**Summary:**

This paper introduces a series of techniques to decouple dynamic objects from monocular videos, which present impressive static background recovery with shadow removal.  How to remove the disturbance from dynamic objects/occluders is a long-standing problem in 3D vision domain. I appreciate the simple but valuable design, i.e., the skewed entropy loss with decoupled NeRF and an extra shadow field. However, many previous methods have adopted similar techniques for foregroud/background content separation. Some related literatures are not included in the paper. Besides, it reads like a CVPR-tyle paper. I'm not sure whether it's suitable for NIPS.

**Questions:**

1.  The skewness hyper-parameter $k$ is critical in practice because it determines the proportion of the `dynamic'. And it's sensitive according to the provided experiments. Is it possible to automatically tune the parameter?
2. The motivation and mechanism of the skew entropy loss is similar to the beta distribution proposed by [1]. As it is regarded one of the major contribution, a thorough comparison and discussion is deserved.
3. The topic is NVS in dynamic environments. Some related literatures are absent[1][2].
4. The shadow NeRF concept has been introduced in [3], which should be included in related literature.

[1] Neural Volumes: Learning Dynamic Renderable Volumes from Images
[2] Neural Scene Graphs for Dynamic Scenes
[3] Shadow Neural Radiance Fields for Multi-view Satellite Photogrammetry

**Ethics Review Area:**

["I don’t know"]

**Limitations:**










**Strengths And Weaknesses:**

Strenghts:
1. The skewed-entropy loss is able to flexibly separate the static and dynamic parts.
2. Separating the time-varying shadows and the static appearace via a shadow field.
2. The decouple performance and NVS quality significantly outperforms SOTA.

---

> ### Author Response · Authors · 2022-08-02
> **Response to PH91**
>
> We thank the reviewer PH91 for the detailed comments and constructive suggestions. Below are our responses to the questions.
>
> ### R4-Q1 Hyperparameter Tuning
> Please refer to All-Q2 for an analysis of the hyperparameter sensitivity and potential approaches to improve the robustness of our method. As suggested in All-Q2, we could rely on ground truth static views to initialize the static component, or we could use those as validation views to automate the tuning process for hyperparameters.
>
> ### R4-Q2 Comparison with Other Sparsity Regularization
> Many losses, including the binary entropy loss in our method, beta distribution loss from [1] and Laplacian distribution loss from [2] all serve a similar purpose: reducing entropy in the variable distribution and promoting sparsity by encouraging values to approach 0 or 1. By appending an exponential function, all of them could be skewed in a similar style to suppress and regularize the dynamic component. We have experimented with the Laplacian distribution loss and have achieved similar decoupling results to the binary entropy loss we used in the paper.
>
> However, a crucial difference between skewed binary entropy loss and other losses is that the former has a vanishing gradient when approaching 0, while the other losses (beta and Laplacian) have gradients approaching infinity near 0. This difference has its pros and cons. The disadvantage is that we have to design and tune an additional loss (eq 10) to remove the low-density floaters in the free space. The benefit is that we potentially allow space that is initially learned as fully static (with $w=0$) to change to dynamic during the optimization.
>
> ### R4-Q3 Additional Literature
> Thank you for the constructive comments on the additional related works, we have added them in the revision.
>
> [1] Neural Volumes: Learning Dynamic Renderable Volumes from Images
>
> [2] LOLNeRF- Learn from One Look

---

> > ### Comment · Reviewer_PH91 · 2022-08-09
> > **Reply**
> >
> > The authors well addressed my concerns. Actually, I really appreciate this paper although the technique is relatively simple.

---

### Official Review · Reviewer_GJS4 · 2022-07-08

**Rating:** 5
**Confidence:** 4
**Soundness:** 3 good
**Presentation:** 3 good
**Contribution:** 3 good

**Summary:**

This paper proposes a new self-supervised approach for segmenting moving objects from the static background. It tackles with issues encountered during training with several techniques, such as skewed entropy, ray regularization, static regularization, and integrating shadow ratio to separate shadows.

**Questions:**

1. I am curious about the robustness of the proposed approach to the hyparameters. See the weakness part.

**Limitations:**

I appreciate the authors's discussions about the limitations of the proposed approach in Sec. 5, which helps the understanding of the suitable scenarios.

**Strengths And Weaknesses:**

## Strengths

1. Originality: the proposed techniques are insightful and are beneficial to the community, such as skewed entropy and ray regularization.
2. Quality: The presented qualitative results are of high quality.
3. Clarity: the manuscript is well-written.
4. Significance: the task of decoupling dynamic objects and static background is important and the proposed approach tackles the long-standing problem with high-quality results.

## Weakness

1. I think the main weakness comes from the scene-level tuning for the hyparameter of k, $\lambda_s$, $\lambda_r$, $\lambda_{\sigma^S}$, $\lambda_\rho$.
2. I understand that to push the number, the author may need grid-search for each scene. However, the lack of analysis for the sensitivity of those hyparameters makes it unclear how robust the proposed approach is.
3. Especially considering in Sec. B of supp, we have 9 sets of parameters for 19 scenes. Even for the same scene (Banana), we have two sets of hyparameters for two tasks (decoupling and novel view).
4. Besides the per-scene tuning results, I recommend the author to report a set of quantitative results with a single set of hyparameters if possible.

---

> ### Author Response · Authors · 2022-08-02
> **Response to GJS4**
>
> We thank reviewer GJS4 for the detailed comments and constructive suggestions. Below are our responses to the raised questions and concerns:
>
> ### R3-Q1 Hyperparameter Sensitivity
> Please refer to All-Q2 for an analysis of the hyperparameter sensitivity and potential approaches to improve the robustness of our method.
>
> For the scene “Banana”, two separate sets of hyperparameters are used to maximize performance in different aspects: scene decoupling and novel view synthesis. We believe the reason why different hyperparameters are needed is the inaccuracy in estimated camera poses, which cause difficulty for the static NeRF network to correctly model some of the high-frequency details in the background. This is also the reason why our method does not consistently outperform the HyperNeRF baseline. However, by slightly reducing the regularization of the dynamic component and allowing it to contain part of the static background, the network is able to abuse temporal dependency to account for the bias caused by incorrect camera poses, achieving better quality in novel view synthesis with a tradeoff to the correctness of decoupling.
>
> ### R3-Q2 Results with Universal Hyperparameters
> Please refer to All-Q3 for the numerical results and analysis of our method with a universal set of hyperparameters.

---

### Official Review · Reviewer_xgPY · 2022-07-10

**Rating:** 7
**Confidence:** 4
**Soundness:** 4 excellent
**Presentation:** 4 excellent
**Contribution:** 4 excellent

**Summary:**

This paper presents D^2NeRF, a self-supervised method that takes a monocular video and learns a 3D scene representation that decouples moving objects, including their shadows, from the static background. In addition, this paper proposes a novel loss to promote the correct separation of phenomena for static and dynamic filed. The authors further propose a shadow field network to detect and decouple dynamically moving shadows. A new dataset was proposed, containing various dynamic objects and shadows. Extensive experiments demonstrate that the proposed method can achieve better performance than state-of-the-art approaches in decoupling dynamic and static 3D objects, occlusion and shadow removal, and image segmentation for moving objects.

**Questions:**

please refer to the weakness. Add more comparison results with exiting methods.
* do not compare with other motion decoupling methods, such as STNeRF[18], NSFF[24] and DynNeRF[11], SIMONe[19], STaR[63], although their experiment setting may be inconsistent with the proposed methods. However, I think in the synthetic dataset, all of them can be reproduced, please add some of their results for a complete comparison.
* It seems to lack generalizability, just train and test in the same dataset.

**Ethics Review Area:**

["I don’t know"]

**Limitations:**

The authors have discussed the limitation, and all of them are a considerable challenge. I have no idea to solve them, and I think it's a trade-off.

**Strengths And Weaknesses:**

Strength:

* New dataset for static and dynamic field decomposition.
* Self-supervised method that takes a monocular video and learns a 3D scene representation that decouples moving objects, including their shadows, from the static background.
* Novel loss to promote the correct separation of phenomena for static and dynamic filed.
* A shadow field network to detect and decouple dynamically moving shadows
* SoTA results in several tasks such as decoupling dynamic and static 3D objects, occlusion and shadow removal, and image segmentation for moving objects.

Weakness:
* require the accurate camera, suffer from high-frequency view-dependent radiance change, which has been discussed.
* do not compare with other motion decoupling methods, such as STNeRF[18], NSFF[24] and DynNeRF[11], SIMONe[19], STaR[63], although their experiment setting may be inconsistent with the proposed methods. However, I think in the synthetic dataset, all of them can be reproduced, please add some of their results for a complete comparison.
* When I refer to the supplementary video, there are still some wrong static and dynamic field decompositions, such as the keyboard. And I think this is a tradeoff.
* It seems to lack generalizability, just train and test in the same dataset.

---

> ### Author Response · Authors · 2022-08-02
> **Response to xgPY**
>
> We thank reviewer xgPY for the detailed comments and constructive suggestions. Below are our comments on the raised questions and concerns:
>
> ### R2-Q1 Additional Comparisons
> We have evaluated NSFF as an additional baseline in background reconstruction and video segmentation tasks. Please refer to All-Q1 in the general response for the results and analysis.
>
> Unfortunately, comparisons to other suggested methods cannot easily be done in a fair setting as they require additional supervision or work in more constrained scenarios: STNeRF [18] targets at dynamic humans and requires extensive supervision including human masks and bounding boxes. DynNeRF [11] similarly requires accurate dynamic masks as training input. Comparison to them will be equivalent to comparing with the pre-trained segmentation networks they use behind the scene, whereas we have already included Motion Grouping [61] as a SOTA motion segmentation method as a baseline. STaR [63] requires multi-view video as training input and only works in scenes with a single rigid dynamic object. SIMONe [19] segments out all objects from the background without distinguishing between dynamic and static ones, and requires multiple sequences on each scene for training. We believe the field of self-supervised 3D motion decoupling is still under-explored and hope our work can motivate additional research in this direction.
>
> ### R2-Q2 Generalizability
> We agree that our method lacks the ability to generalize across scenes and requires time-consuming per-scene optimization. To the best of our knowledge, this is a limitation in all existing temporal-dependent NeRF-based methods. While extracting shared priors across datasets can certainly improve method efficiency, trading-off with the reconstruction robustness on scenes with a large variety of different motion, geometry and appearance is a non-trivial task and we leave that as a potential direction for future research.

---

### Official Review · Reviewer_Xgiz · 2022-07-13

**Rating:** 7
**Confidence:** 5
**Soundness:** 4 excellent
**Presentation:** 4 excellent
**Contribution:** 3 good

**Summary:**

This paper proposes a method to segment and decouple dynamic objects while recovering the static environment from a monocular video. It adapts from NeRF and its extension Hyper NeRF but with improved handling of shadow regions as well as a loss to promote correct separation of dynamic and static regions. It demonstrates plausible motion segmentation and shadow removal result compared to recent NeRF based methods.



**Questions:**

see the questions in "strength and weakness"

**Ethics Review Area:**

["I don’t know"]

**Limitations:**

The method cannot handle high frequency view-dependent radiance change due to the monocular moving camera setting.

**Strengths And Weaknesses:**

This paper is a solid development for using NeRF to reconstruct from monocular videos with dynamic foregrounds. The paper in general is well-written and the claims are well supported. The adaptation it made to handle shadow and foreground & background separation is elegant and seems to be effective. It definitely holds value to people working on similar problems and deserves publication.

There are a couple of things I hope the authors could comment on.

(1) In Ln. 226, “To demonstrate the ability of fully self-supervised scene decoupling, we do not apply any masks when registering real-world images using COLMAP”. In my experience, without feeding masks, COLMAP tends to make wrong estimation of camera poses when the foreground is sufficiently large, which will definitely results in wrong reconstruction of the static background. It seems the proposed do not attempt to update the camera pose during optimization, so the claim above looks confusing to me.

(2) It would be nice if the author also visualizes the depth map of the reconstructed foreground / background.

(3) The results of hyperNeRF looks far worse compared to the proposed method even in the dynamic region. Given that the proposed method seems to have little difference to hyperNeRF at least in the dynamic regions, the current result surprises me a bit. Could the author make additional comments on the possible reasons?

---

> ### Author Response · Authors · 2022-08-02
> **Response to Xgiz**
>
> We thank reviewer Xgiz for the detailed comments and constructive suggestions. Below are our responses on the raised questions and concerns:
>
> ### R1-Q1 COLMAP Mask
> We wish to demonstrate the performance of our method when applied to scenes containing challenging dynamic occluders that cannot be easily masked out using a pre-trained semantic segmentation network, for example, liquid pouring down to a different container. Therefore, we do not apply any masks in any stage of the training, including the camera registration stage for our real-world scenes.
>
> For our real-world captures, the background contains sufficient details and the scale of dynamic occluders is small enough, allowing the majority of features to be extracted and matched in the background. This ensures camera registration via COLMAP is done with a reasonable level of accuracy. The mean reprojection errors typically range from 0.49px to 0.73px in our dataset.
>
> ### R1-Q2 Depth Map
> Thank you for the constructive suggestion, we have added depth maps for the decoupled dynamic/static and both components together to the supplementary to showcase the correctness of the geometry reconstructed.
>
> ### R1-Q3 HyperNeRF Result
> We have empirically found that HyperNeRF suffers from reconstructing dynamic objects for scenes with rapid object motion near the static background (Balloon, Water, Pick, Broom, etc). The simple MLP in HyperNeRF cannot easily represent the sharp discontinuities in the deformation field, hence strongly biasing the scene towards smooth motion. While such temporal smoothness is desirable within the dynamic objects and does not cause issues for dynamic objects that are far away from any static geometry, it becomes a limitation for scenes where dynamic and static geometry are close to each other, for example, when a dynamic broom quickly sweeps above a static floor. Meanwhile, our method avoids both of the above issues by simply decoupling the scene and allowing HyperNeRF to focus on the geometry, appearance and deformation of the dynamic objects only.

---

### Author Response · Authors · 2022-08-02
**General Response**

We thank reviewers for the detailed and considerate feedback. Below are our responses to the common questions.

### All-Q1 Additional Results

As suggested, we have evaluated NSFF under the background reconstruction and video segmentation tasks and report the additional comparison in the tables below. We disabled the hard mining initialization via pre-trained 2D masks in NSFF and trained it for 500k on each scene. However, please note that although NSFF similarly learns separated dynamic and static NeRFs, it is designed to only maximize the quality of view synthesise rather than scene decoupling and does not incorporate any regularization on the dynamic NeRF. While it consistently reaches PSNR between 24 to 31 in the novel view synthesis, its decoupling is extremely unstable and can sometimes lead to completely empty static NeRF being learned.

|  |  | Car |  |  | Cars |  |  | Bag |  |  | Chairs |  |  | Pillow |  | Mean |  |  |
| --- | --- | --- | --- | --- | --- | --- | --- | --- | --- | --- | --- | --- | --- | --- | --- | --- | --- | --- |
|  | LPIPS↓ | MS-SSIM↑ | PSNR↑ | LPIPS↓ | MS-SSIM↑ | PSNR↑ | LPIPS↓ | MS-SSIM↑ | PSNR↑ | LPIPS↓ | MS-SSIM↑ | PSNR↑ | LPIPS↓ | MS-SSIM↑ | PSNR↑ | LPIPS↓ | MS-SSIM↑ | PSNR↑ |
| NeRF-W |  .218 |  .814 | 24.23 |  .243 |  .873 | 24.51 |  .139 |  .791 | 20.65 |  .150 |  .681 | 23.77 |  .088 |  .935 | 28.24 | .167 | .819 | 24.28
NSFF | .200 | .806 | 24.90 | .620 | .376 | 10.29 | [.108] | .892 | 25.62 | .682 | .284 | 12.82 | .782 | .343 | 4.55 | .478 | .540 | 15.64
NeuralDiff | [.065] |  .952 | 31.89 |  .098 |  .921 | 25.93 |  .117 |  [.910] | [29.02] |  .112 | **.722** | 24.42 |  .565 |  .652 | 20.09 | .191 | .831 | 26.27
Ours  | **.062** | **.975** | **34.27** | **[.090]** | **[.953]** | **[26.27]** | **.076** | **.979** | **34.14** | **.095** |  [.707] | **24.63** | **[.076]** | **[.979]** | **[36.58]** | **.080** | **.919** | **31.18**
Ours*   | .103 | [.961] | [33.23] | **[.090]** | **[.953]** | **[26.27]** | .254 | .821 | 25.94 | [.108] |  .703 | [24.57] | **[.076]** | **[.979]** | **[36.58]** | [.126] | [.883] | [29.32]


|  | Car |  | Cars |  | Bag |  | Chairs |  | Pillow | |Mean |  |
| --- | --- | --- | --- | --- | --- | --- | --- | --- | --- | --- | --- | --- |
|  | $\mathcal{J}$↑ | $\mathcal{F}$↑ | $\mathcal{J}$↑ | $\mathcal{F}$↑ | $\mathcal{J}$↑ | $\mathcal{F}$↑ | $\mathcal{J}$↑ | $\mathcal{F}$↑ | $\mathcal{J}$↑ | $\mathcal{F}$↑ | $\mathcal{J}$↑ | $\mathcal{F}$↑ |
MG |  .603 |  .743 |  .363 |  .474 |  [.629] |  [.738] |  [.484] |  [.613] |  .044 |  .080 |  .424 |  .529
NeRF-W |  .072 |  .132 |  .098 |  .162 |  .027 |  .052 |  .154 |  .254 |  .194 |  .314 |  .109 |  .183
NSFF | .083 | .152 | .058 | .104 | .102 | .182 | .046 | .087 | .104 | .188 | .079 | .143
NeuralDiff |  .806 |  .891 |  .508 |  .578 |  .080 |  .144 |  .368 |  .513 |  .097 |  .177 |  .372 |  .461
Ours     |  [.848] |  [.917] |  **[.790]** |  **[.874]** | **.703** | **.818** | **.551** | **.687** | **[.693]** | **[.818]** | **.717** | **.822**
Ours*    |  **.863** |  **.926** |  **[.790]** |  **[.874]** | .038 | .071 | .303 | .427 | **[.693]**| **[.818]** | [.537] | [.623]

Quantitative evaluations on background reconstruction (above) and video segmentation (below). **Best** and [second best] results are highlighted. "Ours*" shows the results of our method with a universal set of hyperparameters (row 6 in Table.3).

### All-Q2 Hyperparameter Sensitivity and Tuning
Due to the self-supervised nature and restrictive monocular video setting, the decoupling currently requires some fine-tuning on challenging scenes with correlated camera and object motion, inaccurate camera parameters, and similar appearance between dynamic objects and background.

The tuning of hyperparameters is indeed crucial in practice. Our provided hyperparameter setting (row 1 in Table. 3) achieves correct decoupling in 10 out of 13 of our real-world scenes and is suitable for the majority of applications, but could still fail and require further fine-tuning under challenging object/camera motion or inaccurate camera parameters.

It would be possible to improve the robustness if additional supervision is provided. Given a sparse set of views on the static background, including additional captures of background at different viewpoints, or simply the video frames where dynamic occluders are absent, the static component could be initialized with those views to improve the model robustness against different hyperparameters and scenes. Besides, if we restrict the applications to scenes containing only common dynamic occluders, we could incorporate a pre-trained semantic image segmentation network to guide the initial stage of optimization.

---

> ### Author Response · Authors · 2022-08-02
> **General Response -- continued**
>
> ### All-Q3 Results with Universal Hyperparameters
> To provide more insights into the robustness of our method, we report the quantitative results of our method with a universal set of hyperparameters in the table above, where “Ours*” uses the hyperparameters described in row 6 in Table.3 for all scenes. With this universal set of hyperparameters, the average performance of our method is within 94% of our best result in background reconstruction (measured in PSNR), and within 76% of our best result in video segmentation (measured in boundary measure $\mathcal{F}$). While the performance on “Car” and “Chairs” fluctuates in a moderate range with different hyperparameters, our method still achieves better results in general compared to the baselines. For the scene “Bag”, our method fails to achieve a correct decoupling with this universal set of hyperparameters. We suspect it is because of the similar appearance between the dynamic object (brown bag) and the background (brown floor and shelf), presenting significant challenges to the decoupling and hence requiring specialised hyperparameters to achieve satisfactory results.

---

### Meta-Review · Area_Chair_epg5 · 2022-09-01

**Recommendation:** Accept
**Confidence:** Certain

**Metareview:**

This paper attacks an interesting problem with NERF, decoupling moving objects, including their shadows, from the static background.  All four reviewers recommend accepting the paper, and the weaknesses identified did not detract from substantive contributions. Therefore I am accepting this paper.

**Award:**

No

---

### Decision · Program_Chairs · 2022-09-14

Accept